# Red GaPAs/GaP Nanowire-Based Flexible Light-Emitting Diodes

**DOI:** 10.3390/nano11102549

**Published:** 2021-09-29

**Authors:** Vladimir Neplokh, Vladimir Fedorov, Alexey Mozharov, Fedor Kochetkov, Konstantin Shugurov, Eduard Moiseev, Nuño Amador-Mendez, Tatiana Statsenko, Sofia Morozova, Dmitry Krasnikov, Albert G. Nasibulin, Regina Islamova, George Cirlin, Maria Tchernycheva, Ivan Mukhin

**Affiliations:** 1High School of Engineering Physics, Peter the Great St. Petersburg Polytechnic University, Polytechnicheskaya 29, 195251 St. Petersburg, Russia; vfedorov.fl@mail.ioffe.ru (V.F.); imukhin@yandex.ru (I.M.); 2Department of Physics, Alferov University, Khlopina 8/3, 194021 St. Petersburg, Russia; mozharov@spbau.ru (A.M.); azemerat@rambler.ru (F.K.); shugurov17@mail.ru (K.S.); moiseev@spbau.com (E.M.); cirlin@beam.ioffe.ru (G.C.); 3Laboratory of Quantum Optoelectronics, National Research University Higher School of Economics, Kantemirovskaya 3A, 194100 St. Petersburg, Russia; 4Centre of Nanosciences and Nanotechnologies, UMR 9001 CNRS, University Paris-Saclay, 10 Boulevard Thomas Gobert, 91120 Palaiseau, France; nuno.amador@c2n.upsaclay.fr (N.A.-M.); maria.tchernycheva@u-psud.fr (M.T.); 5Department of Chemistry, ITMO University, Lomonosova 9, 197101 St. Petersburg, Russia; statsenko@scamt-itmo.ru (T.S.); sofiionova@yandex.ru (S.M.); 6N.E. Bauman Moscow State Technical University, 2nd Baumanskaya str. 5/1, 105005 Moscow, Russia; 7Skolkovo Institute of Science and Technology, Bolshoy Boulevard 30/1, 121205 Moscow, Russia; d.krasnikov@skoltech.ru (D.K.); A.Nasibulin@skoltech.ru (A.G.N.); 8Department of Chemistry and Materials Science, Aalto University, FI-00076 Espoo, Finland; 9Institute of Chemistry, Saint Petersburg State University, Universitetskaya Emb. 7/9, 199034 St. Petersburg, Russia; r.islamova@spbu.ru

**Keywords:** flexible LED, GaPAs, nanowires, single-walled carbon nanotubes, molecular beam epitaxy

## Abstract

We demonstrate flexible red light-emitting diodes based on axial GaPAs/GaP heterostructured nanowires embedded in polydimethylsiloxane membranes with transparent electrodes involving single-walled carbon nanotubes. The GaPAs/GaP axial nanowire arrays were grown by molecular beam epitaxy, encapsulated into a polydimethylsiloxane film, and then released from the growth substrate. The fabricated free-standing membrane of light-emitting diodes with contacts of single-walled carbon nanotube films has the main electroluminescence line at 670 nm. Membrane-based light-emitting diodes (LEDs) were compared with GaPAs/GaP NW array LED devices processed directly on Si growth substrate revealing similar electroluminescence properties. Demonstrated membrane-based red LEDs are opening an avenue for flexible full color inorganic devices.

## 1. Introduction

Nowadays, liquid crystal displays (LCDs) are still dominating in screen technologies, since they gradually displaced bulky cathode ray tubes in the 2000s [1,2,3]. However, in the last few years, organic light-emitting diode (OLED) displays have grown rapidly [4,5], following 30 years of intensive material and device development and heavy investment in advanced manufacturing technologies [6,7,8,9,10,11,12]. Despite the significant progress, OLEDs are still inferior to inorganic materials in terms of luminance, efficiency, and long-term stability, especially in blue and red spectral regions. Therefore, OLED technology has a limited application area, and the most attractive usage revealing advantages of organic materials is flexible devices for wearable flexible applications. Indeed, rigid III-V LED and LCD technologies are difficult to adapt for deformable devices, so OLED solutions are preferable. Possible flexible display and lighting applications are numerous and very diverse [13], including: medical devices for light-based treatments [14], wearable health monitors [15], and an abundance of potential uses in the fashion industry [16], etc.

However, there are some specific application fields, where OLEDs fail to fully satisfy the specifications [3]. For example, augmented reality (AR) devices require fast, very bright and compact light sources. Inorganic microLEDs fulfill these requirements [17,18,19], and provide an additional transparency functionality, tremendously simplifying the optical configuration [20,21]. Another important display application preferable for inorganic materials are wristbands, requiring flexibility, high luminance to guarantee sunlight readability, and a good off-axis performance. Though III-V microLED displays are probably the best candidate, they are still under development [20,21] because of the complexity of a commercial device fabrication. III-V microLEDs are also the best solution for vehicle displays, i.e., central cluster panels and head-up displays, requiring a very high reliability and sunlight readability for the driver safety.

In spite of this high demand, III-V microLED technology still requires development to be implemented in practical applications. Among the most important issues of flexible III-V microLED device fabrication are the transfer to a flexible substrate of the synthesized structures and the application of flexible transparent conductive contacts. III-V technology is mostly based on thin films, which demands complex growth and post-growth procedures to fabricate flexible devices [20,22], while conventional transparent contacts such as ITO have limited flexibility or low performance and stability (e.g., poly(3,4-ethylenedioxythiophene) polystyrene sulfonate (PEDOT:PSS)). Therefore, an alternative approach for III-V microLEDs was introduced: nanowire (NW) array devices [23,24]. Benefiting from the high substrate-independency of the NW geometry, flexible [25] and stretchable [26] microLED devices were demonstrated. While for thin film microLEDs the practical application of large area devices is compromised by the necessity of complex post-growth structuring, polydimethylsiloxane (PDMS)/NW membranes of several square inches area were fabricated using a very inexpensive and simple processing [27]. The PDMS/NW membranes provide good mechanical and material performance, and single-walled carbon nanotube (SWCNT) films can serve as excellent highly stretchable transparent electrodes [28] to these membranes. Moreover, expensive substrates used for synthesis of NW arrays can be reused after transfer, saving valuable materials and reducing fabrication costs [29]. Thus, III-V NW/PDMS membrane LEDs with SWCNT electrodes is a promising technology for highly stretchable, semitransparent, light-weight LEDs of large area for future components of miniLED displays. Taking into account the recent progress in blue and green membrane LEDs [24,30,31], flexible LEDs in the red spectral region is the crucial point of the technology.

Here, to the best of our knowledge for the first time we fabricate flexible red GaPAs/GaP NW/PDMS membrane LEDs. The flexible LED was first designed on the basis of numerical simulations and fabricated following previously reported methods [26,27] which were adapted to the challenging geometry of self-catalyzed NWs grown on Si substrates. For comparison, we also studied GaPAs/GaP NWs-based LEDs before array release from the growth Si substrate.

## 2. Results and Discussion

### 2.1. Modeling of GaPAs/GaP NW LEDs

The optimal material composition and geometry parameters of the emitters and active region were defined by the calculation of axial p-i-n GaPAs/GaP NW LED heterostructure. We studied the device function using a drift-diffusion model taking into account the Fermi distribution of electrons and holes, and the Shockley-Read-Hall and radiative recombination. Based on literature data for the parameters of GaP, GaAs, and their solid solutions, we modeled GaP_x_As_1-x_. The calculation model and the most relevant modeling results are presented in Appendix A (refer Appendix A contain used values for effective masses). Modeling was performed for the fixed doping level of the emitters and the active region, 10^18^ and 10^15^ cm^−3^, respectively.

We used the developed numerical model for axial p-i-n GaPAs/GaP NW heterostructure to evaluate the dependency of the internal quantum efficiency (IQE) and power efficiency (PE) of electroluminescence (EL) signal on the material composition (Figure 1a,b) and lengths of the radiative GaPAs insertion and emitter segments (Figure 1c,d). This evaluation allows defining the optimal NW LED design. The EL conditions are considered as the low pumping regime, i.e., the radiative recombination occurs from the lowest energy level, so the photon energy is almost equal to the band gap energy. Efficiency map in Figure 1a demonstrates that the range of P content in the emitters, corresponding to high efficiency of NW LED, is relatively broad. At the same time, the IQE shows the maximum when the P molar composition of the emitters exceeds by at least 20–30% that of the active region insertion. We associate this value with the minimal value required for an efficient carrier localization in the GaPAs insertion, which prevents carriers from passing over the active region. The insets in Figure 1a show energy band diagrams for two typical situations in which the bandgap of the insertion is lower (respectively, higher) than that of the emitters, explaining the conclusion from the Figure 1a that the IQE is above 90% for the higher P content in the emitters.

However, excessive P content contrast between the emitters and the insertion leads to dramatic thermalization losses, i.e., the energy lost by the carriers, when they slide to the fundamental energy level in the active region. Therefore, the maximum power efficiency (PE) for the P content in emitters below 45% is achieved at 20% P content contrast (i.e., for GaP_<0.45_As_>0.55_ in the emitters and GaP_<0.25_As_>0.75_ in the active region insertion) (Figure 1b). However, in the case of high P content (x > 0.4) in the emitters, the preferable composition of the active region lies in the range of GaP_0.25_As_0.75_–GaP_0.45_As_0.55_ due to the change of the main electronic valley. The optimal composition is defined by the fitting Equations:$$x_{em}=4x_{ins}-0.58\;(\mathrm{for}\;\mathrm{the}\;\mathrm{higher}\;\mathrm{part})\;x_{em}=0.8x_{ins}+0.24\;(\mathrm{for}\;\mathrm{the}\;\mathrm{lower}\;\mathrm{part})$$
there $x_{em}$ and $x_{ins}$ are the P content in the emitters and in the active region insertion, respectively. The conclusion from the Figure 1b is that the optimal $x_{ins}$ corresponding to the $x_{em}=1$ (pure GaP emitters) lies in the range of (0.35–0.4), providing the LED emission in the red spectral range. In this case, there is no optical absorption in the emitter layers. Moreover, electron confinement prevents the injection of minor carriers to emitters, which suppresses the undesirable nonradiative recombination in the emitters.

Considering numerical modeling, the optimal parameters of red NW LEDs lie in the following ranges: the active region insertion length of 80–200 nm and composition of GaP_0.2<x<0.4_As_0.8>1−x>0.6_; doping level of the emitters and the active region, 10^18^ and 10^15^ cm^−3^, respectively; the total emitter lengths of less than 4 µm, while the composition can be chosen as pure GaP, which is preferable for the NW synthesis convenience.

### 2.2. GaPAs/GaP NW LED Array Synthesis

In accordance with the results of numerical calculations, axially heterostructured GaPAs/GaP NWs used in this study were grown on p-type (10^20^ cm^−3^) Si (111) substrates in a solid-source molecular beam epitaxy (MBE) Veeco GEN-III setup via the self-catalyzed vapor–liquid–solid (VLS) growth mechanism using Ga as a catalyst. Prior to the growth, Si substrates were treated with the modified Shiraki cleaning procedure [32], finalized with a formation of a thin surface oxide layer in boiling ammonia-peroxide water mixture (1:1:3). To promote formation of pinholes in the oxide layer required for vertical NW growth, substrates were thermally annealed in the MBE chamber for 30 min at a temperature of 780 °C [33].

NW growth was initiated by simultaneous opening of Ga and P_2_ shutters. Ga-cell beam equivalent pressure (BEP) was kept constant at 8×10^−8^ Torr throughout the growth and corresponded to a 3.17 nm/min GaP planar growth rate. The growth of n-GaP/i-GaPAs/p-GaP axial NW heterostructures was performed via controlling the dopant fluxes and the growth temperature: to form n- and p-doped emitters, top and bottom NW segments were intentionally doped with Be (T_Be_ = 780 °C) and Si (T_Si_ = 1200 °C) using standard effusion cells, respectively [27,34]. The NW growth was initiated with a synthesis of p-type doped GaP segments with a height of 1.2–1.5 µm at V/III BEP ratio of 12 and the elevated growth temperature of 640 °C, which prevents catalytic droplet inflation due to high solubility of Be in Ga [35]. The estimated doping level is 10^18^ cm^−3^. In further growth stages, the substrate temperature was reduced by 10 °C in order to obtain stable formation of the GaPAs insertion and n-type GaP segment.

A thin ~100–150 nm GaP_0.35_As_0.65_ intrinsically doped segment was grown between GaP emitters using the procedure described in [36]. This design corresponds to the high efficiency of LED in accordance with numerical calculations. To obtain the required chemical composition of the GaP_0.35_As_0.65_ insertion, the As/P BEP ratio was set to six. During the GaPAs insertion growth, the cumulative As and P to Ga ratio was set to 42; it should be noted that this value also prevents catalytic droplet inflation [36]. Group V fluxes were switched by closing/opening both molecular beam shutters and cracking source needle valves. Our previous study on the formation of GaPAs segments in GaP NWs under identical growth conditions [36] shows that the GaPAs segment diameter is determined by VLS growth front area and equal to the GaP NW underneath.

Top GaP NW segments (height ~1.5 µm) forming n-emitters were grown with a V/III beam equivalent pressure ratio of 18. Finally, NW growth was interrupted by closing both Ga and P_2_ shutters and subsequent sample cooling. An in situ analysis of the NW crystal structure performed with reflection high energy electron diffraction (RHEED) patterns demonstrated that the described growth conditions yielded stable formation of GaP NWs with a zinc-blende structure as it was previously reported in [36,37,38].

Representative scanning electron microscopy (SEM) (Carl Zeiss SMT AG Company, Oberkochen, Germany) images of the synthesized p-i-n GaPAs/GaP NW array samples on Si, obtained with a Zeiss SUPRA 25 SEM setup, are shown in Figure 2. The NW array has about 0.11 NW/µm^2^ density (Figure 2a), 100 nm average wire diameter and 3–4 µm height (Figure 2b). The morphology of NW arrays was rather inhomogeneous that is typical for a self-catalyzed growth on non-patterned substrates. It can be noted that the self-catalyzed NW array presents irregular morphology due to the substrate silicon surface oxide layer non-uniformity [39]. Depending on the NW nucleation condition droplet consumption can occur for part of NWs leading to the interruption of growth and formation of bulge-shaped NW tips.

Photoluminescence spectra of the synthesized NW demonstrated main line position at 650 nm (see Appendix A for details, Appendix A shows photoluminescence data), which confirms the composition of GaP_0.35_As_0.65_ insertions. We also observed a parasitic 3D growth resulting in a formation of 0.5–1 µm sized GaP particles on the substrate. However, these 3D particles do not participate in the LED on Si substrate, nor in the flexible LED operation, since the particles were buried in SU-8 and not transferred to PDMS/NW membranes after release (see device fabrication section for the details).

### 2.3. Sylgard Preparation

For the polymer membrane formation, a commercial PDMS (Dow Corning Sylgard 184, Midland, MI, USA) was mixed in a base to a curing agent ratio of 10 to 1.

### 2.4. Fabrication of the Reference NW LED on Si Substrate

At the first step we processed GaPAs/GaP NW array LED devices directly on Si growth substrate (denoted further as ‘LEDs-on-Si’) in order to test their performance on the rigid substrate and compare it with flexible PDMS/NW membranes released from the substrate. Firstly, p-contact was made via a thin Al layer deposition on the back side of the Si substrate and consequent rapid annealing at 300 °C temperature in the N_2_ atmosphere. The NWs on Si were encapsulated into epoxy-based photoresist SU-8 to provide planarization for further transparent top indium tin oxide (ITO) contact application. This process was carried out as follows. The SU-8 layer was spin-coated onto the sample surface to fill the volume between NWs and cover NW tips. After the ultraviolet (UV) exposure of photoresist, the sample was annealed at 95 °C on a hot plate for 5 min and then at 200 °C for 10 min to cure the SU-8 layer. Release of NW tips buried in SU-8 was achieved iteratively by O_2_ plasma etching and SEM control. Then, 5 mm^2^ of ITO contact mesas of 100 nm thickness were deposited to the n-doped segments of NWs via magnetron sputtering through the mask with use of BOC Edwards Auto 500 system (Edwards, Burgess Hill, UK). The workflow schematic is presented in Figure 3.

### 2.5. Fabrication of Flexible PDMS/NW LED

In order to encapsulate 4 µm long GaPAs/GaP NWs into PDMS we used the G-coating method suitable for a thin PDMS membrane fabrication [34]. For further electrical contacting, the tips of NWs were opened via O_2_/CF_4_ plasma etching, thus achieving a homogeneous thickness of PDMS membranes, about 3 µm thick, so the highest parts of NWs were revealed above the membrane surface (Figure 4a). Transparent flexible contact pads to the top n-doped segments of NWs (Figure 4a) were based on SWCNT films [40,41], the size of square contact pads was about 1 mm^2^. The SWCNTs were synthesized by the aerosol (floating catalyst) CVD method described in detail elsewhere ([42,43]) and downstream of the reactor in the form of randomly oriented nanotube film on microporous nitrocellulose filters. After the top contact application, the PDMS/NW membranes were mechanically peeled from Si substrate with the help of a razor blade. The peeled membranes were flipped for a bottom contact application, then again 1 mm^2^ SWCNT contact pads were applied to the PDMS/NW membranes directly opposite to the previously applied top contact pads (Figure 4b). As a result, the PDMS/NW membranes had several areas contacted with SWCNT on both sides of the membrane. For electrical measurements small silver lacquer droplets were applied to the SWCNT contact pads (Figure 4c). A photo of a representative flexible LED is presented in Figure 4d.

Few samples possessed small (about 100 µm in diameter) tears in the PDMS membrane. Although it can be expected that an LED can be electrically shunted through these tears, we have never observed shunts even when the SWCNT films were put directly on a tear, probably because the 3 µm thickness of the PDMS membrane is sufficient to avoid undesirable penetration of the SWCNT film through these holes.

It is worth noting that PDMS/NW membranes were also released using removable cap film providing mechanical support for the membrane. In practice, thin PDMS/NW membranes are very fragile for a manual mechanical peeling, so the release procedure requires gentle operations, and microscale damages of the membranes can appear during the release process. A strong mechanical support of a thick cap film can significantly facilitate the release procedure. Meanwhile, the supporting cap film needs to be removable; indeed, for future applications of LED membranes in multi-color devices (e.g., in combination with blue or green color light-emitting membranes [24,44]), after the PDMS/NW membrane release, the cap film material should be removed for further processing. In our work, we suggested to use polyurethane (PU) as a promising candidate for cap film material, since it can be dissolved in dimethylformamide (DMF) without critical degradation of PDMS membrane material, while III-V NWs and SWCNT films are chemically inert to DMF (see Appendix A for details, Appendix A demonstrate PU synthesis scheme).

### 2.6. Electroluminescence Characterization

The fabricated samples of LEDs-on-Si and flexible LEDs were tested in a probe station using a Keithley 2400 sourcemeter (Keithley Instruments, Solon, OH, USA). In the case of LEDs-on-Si, the upper contact connection was performed with a probe directly positioned on the ITO contact pad, the bottom contact connection was made to the metallized Si substrate with the conductive table of the probe station. The flexible LED samples were contacted in a similar manner: the upper contact connection was performed with a probe tip touching silver lacquer droplet on a SWCNT contact pad of the top side of the flexible LED, the bottom contact connection was to the corresponding opposite SWCNT contact pad (also furnished with a silver lacquer droplet) lying on a special Au/Ge metal film contacted with another probe. In this configuration of electric connection, we minimized the effect of possible parasitic electrical barriers. It should be noted that flexible LEDs cannot be bonded by a standard method with metallic wires, e.g., in a bonding machine or with a deposition of metallic contact bars to a remote contact pad, since the PDMS softness and temperature regime prohibit contact annealing for higher than 250 °C for PDMS and 100 °C for SWCNT (these values are taken from our tests). Therefore, some flexible devices based on PDMS/NW membranes exhibit significant parasitic electrical barriers in I-V curves and electron beam induced current maps [45], and electrical characterization of NW/PDMS membranes requires careful experiment preparation and result data analysis.

EL measurements were carried out at room temperature in the same probe station setup at a constant voltage applied to LED samples. EL spectra were acquired with a AvaSpec-ULS2048XL-EVO-RS spectrometer (Avantes, Louisville, CO, USA), and a monochromator Horiba FHR1000 with a cooled CCD matrix Horiba Symphony for LEDs-on-Si and flexible LEDs, respectively.

### 2.7. LEDs-on-Si Characterization

Figure 5a illustrates a current-voltage (I-V) curve of a representative LEDs-on-Si that exhibits a rectifying behavior at forward bias. High opening voltage of about 5 V indicates the presence of parasitic potential barriers. These barriers at both Si substrate/NW and NW/ITO interfaces lead to a reversed Schottky diode characteristic [34,46], limiting the LED performance. The calculated current density corresponding to an individual ITO contact pad (mesa) is estimated at 250 A/cm^2^ (20 nA/NW) at 8 V forward bias. A typical EL spectrum measured at 8 V is shown in Figure 5b. The main EL line lies at 650 nm, which corresponds to the red spectral range.

### 2.8. Flexible LED Characterization

Similar to LEDs-on-Si, the measured I-V curves for flexible LEDs demonstrated rectifying behavior in both forward and reverse bias (Figure 6a). It can be explained by Schottky nature of the SWCNT contact to n-GaP, which were studied and proven in [34]. The parasitic Schottky barrier is undesirable for a high LED performance. One possible solution to this problem can be a pre-deposition of a thin metallic layer to the n-GaP emitters, effectively reducing potential barrier to n-GaP material: a similar method was used for p-GaN shell of InGaN/GaN blue and green NW LEDs, covered with 5/5 nm Ni/Au [47,48]. After metallization, the NWs can be contacted with SWCNT or other conductive materials without forming a Schottky barrier [24,26]. The search for a proper thin metallic layer composition for ohmic contact to n-GaP will be the subject of our future studies.

The knee voltages were found to be about 7 and 2.5 V for the forward and reverse bias, respectively. It should be noted that the reversed opening voltage is lower than for the LEDs-on-Si (about 7.5 V), which means that SWCNT/p-GaP contact has a lower parasitic potential barrier in comparison with the p-GaP/Si substrate interface.

The fabricated flexible LEDs demonstrated EL in the red spectral range (inset in Figure 6a). The EL spectra were measured at 8 V of forward bias (0.045 mA current), a representative spectrum is shown in Figure 6b. The main EL line is positioned at 670 nm wavelength, corresponding to the direct bandgap transition of GaP_1−x_As_x_ with x = 0.35 content [49]. A rather broad full width at half maximum (FWHM) can be attributed to the inhomogeneous distribution of NW parameters, including non-uniform As content in the GaPAs segment [36]. The inhomogeneous distribution of NW arrays can also explain the 20 nm redshift of the flexible LED EL peak position in comparison with the LEDs-on-Si EL (Figure 5b). Note that 3D GaP islands (see Figure 2) were fully buried into the polymer matrix and were not contacted by the top SWCNT film contact. This prevented the direct injection of carriers into 3D islands and, thus, they did not affect the EL signal.

The flexible LED operation current density was roughly estimated at 50 A/cm^2^ (4 nA/NW), i.e., 5 times lower than for the LEDs-on-Si. It should be noted that for a stable LED operation the current density value at 5–10 A/cm^2^ is preferable since small diameter GaPAs/GaP NWs embedded into a PDMS membrane can suffer overheat problems due to the low heat capacity of small diameter NWs and bad thermal conductivity of the PDMS matrix, and the Joule heating can destroy NWs [50]. To achieve the improved performance, flexible LEDs based on GaPAs/GaP NW/PDMS membranes require further development of both work current density decreasing and improved heat evacuation. Meanwhile, the applications require a high LED power, which can be achieved at low current density by an increase in the NW array density.

During EL measurements the NWs were degrading because of the high Joule heating and weak heat evacuation preventing a long measurement experiment. Effective heat management can be achieved by lowering series resistance (including parasitic Schottky barriers), improving heat conductive properties of contacts or PDMS matrix by adding fillers such as diamond nanoparticles [51]. Another solution can be the application of impulse instead of constant voltage, however, it will lead to a decrease in LED intensity, which can be undesirable for highly bright purposes (i.e., AR). Flexible LED breakdown caused by overheating did not allow us to study EL dependence on applied current.

## 3. Conclusions

In conclusion, we developed and demonstrated flexible red LEDs based on vertical GaPAs nanowire arrays embedded in a PDMS membrane and contacted with transparent SWCNT films. Axial geometry of GaPAs/GaP NW LEDs was studied numerically and successfully synthesized via MBE growth. We compared experimental flexible LED EL performance with fabricated LEDs-on-Si (based on the same vertical GaPAs nanowire array samples and processed directly on the Si growth substrates, encapsulated into SU-8 epoxy resin and contacted with ITO pads). Both LEDs-on-Si and flexible LEDs have dominant EL lines at 650 and 670 nm, respectively. The measured current densities at working condition appeared to be 5 times lower for flexible LEDs in comparison to LEDs-on-Si (50 and 250 A/cm^2^, respectively), while the working voltage is similar for both flexible LEDs and LEDs-on-Si with an 8 V value.

Further development of flexible LEDs should be aimed at the decreasing of working current density and improvement of heat evacuation leading to a low device lifetime due to the overheat degradation. Improvement of the heat management and strategies to lower working current densities will be the subject of further studies.

The demonstrated flexible red LEDs can be employed for the development of light-weight transparent full color displays for future applications in wearable, biomedical, and vehicle-integrated devices.

## Figures and Tables

**Figure 1 nanomaterials-11-02549-f001:**
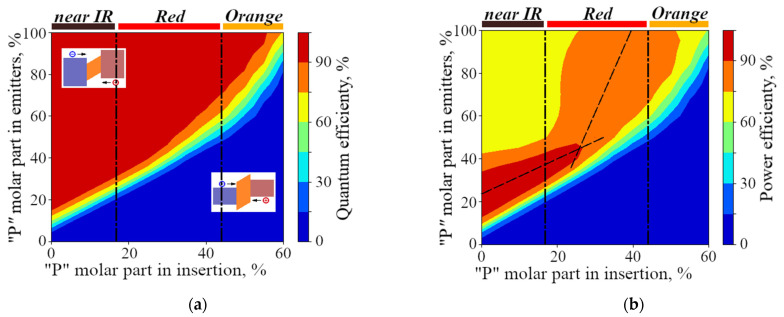
Calculated maps of internal quantum efficiency (IQE) and power efficiency (PE) dependent on P content in emitters and active region insertion (**a**,**b**), and nanowire (NW) (**c**,**d**) length display IQE and PE maps, respectively, dependent on the total NW length and the insertion length. For the lengths of NWs less than 5 µm the dependency on both lengths is negligible for the insertion length lying in the range of (80–200) nm.

**Figure 2 nanomaterials-11-02549-f002:**
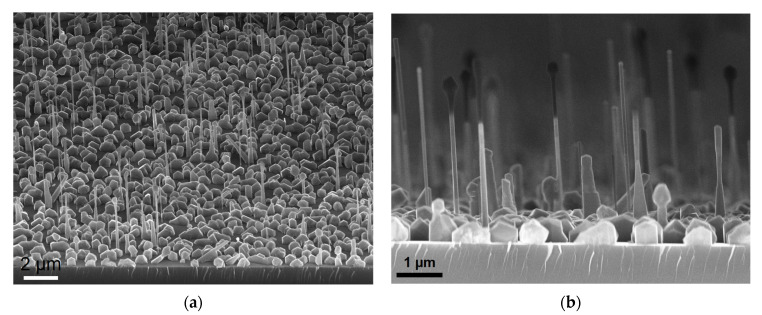
SEM images of a representative as-grown GaPAs/GaP NW light-emitting diode (LED) array in a 45° (**a**) and side view (**b**).

**Figure 3 nanomaterials-11-02549-f003:**
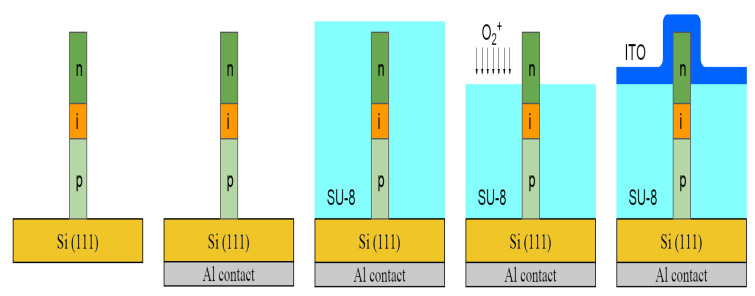
Workflow schematic for NW LED-on-Si.

**Figure 4 nanomaterials-11-02549-f004:**
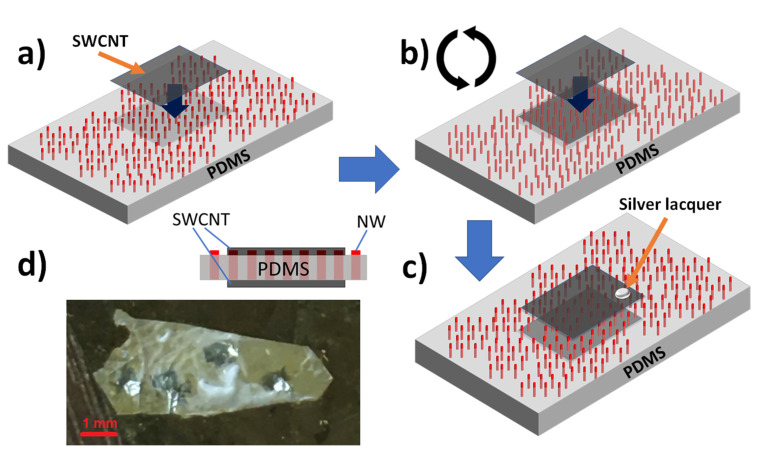
(**a**–**c**) Workflow schematic of single-walled carbon nanotubes (SWCNT) contact application for flexible LED, (**d**) photo of a representative flexible LED. The insert schematically shows the cross-section of the membrane NWs-based LED.

**Figure 5 nanomaterials-11-02549-f005:**
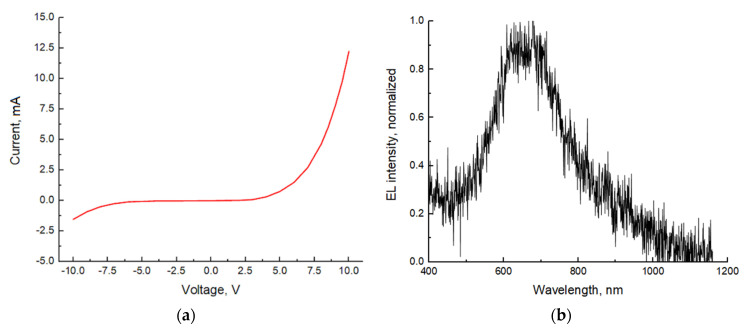
I-V curve of an individual ITO contact pad (**a**) and electroluminescence (EL) spectrum (**b**) of a representative LED-on-Si.

**Figure 6 nanomaterials-11-02549-f006:**
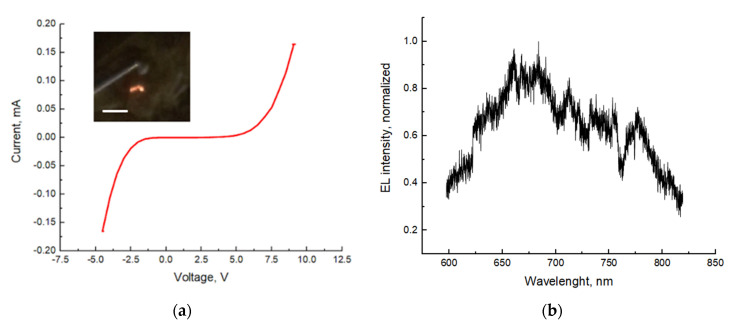
I-V curve (**a**) and EL spectrum (**b**) of a representative flexible LED. The inset in (**a**) demonstrates a photo of the working device, the scale bar is 3 μm.

## Data Availability

The data presented in this study are available on request from the corresponding author. The data are not publicly available due to the author’s readiness to provide it on request.

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
