# Peer review of "Red GaPAs/GaP Nanowire-Based Flexible Light-Emitting Diodes"

_nanomaterials, 2021, doi:10.3390/nano11102549_

Round 1

Reviewer 1 Report

The manuscript presents some interesting advances in flexible LEDs based on nanowires. The aim is to study red LEDs based on GaP nanowires including a GaAsP segment as the active region.

There are a few issues that must be addressed before the paper can be accepted. It may be that there is a mix-up with the notations, but the discussion around figure 1 (line 97-121) needs to be improved and clarified.

  • I can only assume that the dashed lines in figure 1 b illustrates the line of optimum power for two areas of the plot. And that this line corresponds to the equation on line 116-117. I hope that this equation is wrong. This does not fit with the line in the figure.
  • The figure illustrates a composition range of the insert of 0-60%, but the conclusion of the optimum insert for GaP is that it should be 65%. Outside the plots of 1 a and 1b.
  • 65% GaAsP is about 600 nm, not red but rather yellow. Should it be 35%?
  • It would be helpful if the position of red was indicated in 1a and 1b, to illustrate the target composition of the inset segment.

It would be useful to mention if there are any issues since GaP is indirect and 35 % P in GaAsP is direct.

The diameter of the segments should be indicated. From figure 2, there seems to be a spread in the diameter as well as the length.

There is a lot of parasitic growth, does it influence the EL signal?

The PL emission is relatively narrow. In the conventionally connected device the line width has increased and in the flexible it is even worse, and the latter spectrum is also quite jagged. This should be addressed.

The inset in figure 6 appears to show a red hotspot. There is no scale here so it is difficult to understand what it illustrates.

The references are in a strange format. Is this acceptable? I would prefer a more conventional numbering.

Finally, though the English is acceptable, it could do with some improvements to make the manuscript more readable.

Reviewer 2 Report

Neplokh et al. demonstrate in this manuscript the fabrication of flexible LEDs based on GaPAs/GaP nanowire ensembles embedded in a polydimethylsiloxane film and emitting in the red spectral region. This achievement is based on related previous work of the same and other groups, as mentioned in the introduction. The main novelty of the current study is the choice of nanowire heterostructures that emit in the red spectral region. The work is described in detail, the manuscript is clearly structured and well written. Thus, I recommend its publication provided the following minor deficiencies are addressed:

-The comprehensive study involves a theoretical component to determine the appropriate As content in the active region. For these simulations, it should be made clear whether they were based on a simplified one-dimensional model or fully three-dimensional calculations were carried out, taking into account, e.g., band bending due to Fermi level pinning at the nanowire sidewalls.

-The description of Fig. 1a refers to „ a maximum when the P molar composition of the emitters exceeds by 20-30% that of the active region insertion.“ I find this description misleading since the quantum efficiency is very high in about half of the diagram, i.e. there is not really a pronounced maximum.

-Likewise, the statement “For the lengths of NWs less than 5 µm the dependency on both lengths is negligible for the insertion length above 140 nm.” is made for both Fig 1c and d but holds only for the latter.

-On what data is the following statement based? “However, insertion length above 500 nm is undesirable, because carrier localization in the active region decreases and the radiative recombination rate becomes less intensive.” The corresponding range of insertion lengths is not shown in Fig. 1.

-Page 8 contains several paragraphs describing a processing variation involving polyurethane, but no results are shown for this approach. Hence, including this description does not provide any benefit to the reader.

-For the LED data presented in Fig. 5 and 6, the device size should be indicated.

-In the conclusions, the authors claim that their LEDs can be employed for the development of high brightness light-weight transparent full color displays. However, these LED suffer from severe limitations, e.g., they have to be operated at a very high current density, likely because of the poor contacts, and their lifetime is very small, which the authors attributed to overheating. Therefore, the authors have to scale back their claim. Likewise, the rather noisy electroluminescence spectrum does not really justify to use the word “bright” in the description of Fig. 6a.

-Finding items in the bibliography is rather cumbersome because in the text, alphanumerical codes are used to identify references, but the order in the bibliography is not alphabetical.

-In the supporting information, it should be “conduction band” instead of “conductivity band”.
